# Detection of Sulfite Dioxide Residue on the Surface of Fresh-Cut Potato Slices Using Near-Infrared Hyperspectral Imaging System and Portable Near-Infrared Spectrometer

**DOI:** 10.3390/molecules25071651

**Published:** 2020-04-03

**Authors:** Xiulin Bai, Qinlin Xiao, Lei Zhou, Yu Tang, Yong He

**Affiliations:** 1College of Biosystems Engineering and Food Science, Zhejiang University, Hangzhou 310058, China; xlbai@zju.edu.cn (X.B.); qinlxiao@zju.edu.cn (Q.X.); zhoulei_17@zju.edu.cn (L.Z.); 2Key Laboratory of Spectroscopy Sensing, Ministry of Agriculture and Rural Affairs, Hangzhou 310058, China; 3College of Automation, Guangdong Polytechnic Normal University, Guangzhou 510665, China

**Keywords:** portable near-infrared spectrometer, near-infrared hyperspectral imaging, sulfite residue, fresh-cut potato slices, sodium pyrosulfite

## Abstract

Sodium pyrosulfite is a browning inhibitor used for the storage of fresh-cut potato slices. Excessive use of sodium pyrosulfite can lead to sulfur dioxide residue, which is harmful for the human body. The sulfur dioxide residue on the surface of fresh-cut potato slices immersed in different concentrations of sodium pyrosulfite solution was classified by near-infrared hyperspectral imaging (NIR-HSI) system and portable near-infrared (NIR) spectrometer. Principal component analysis was used to analyze the object-wise spectra, and support vector machine (SVM) model was established. The classification accuracy of calibration set and prediction set were 98.75% and 95%, respectively. Savitzky–Golay algorithm was used to recognize the important wavelengths, and SVM model was established based on the recognized important wavelengths. The final classification accuracy was slightly less than that based on the full spectra. In addition, the pixel-wise spectra extracted from NIR-HSI system could realize the visualization of different samples, and intuitively reflect the differences among the samples. The results showed that it was feasible to classify the sulfur dioxide residue on the surface of fresh-cut potato slices immersed in different concentration of sodium pyrosulfite solution by NIR spectra. It provided an alternative method for the detection of sulfur dioxide residue on the surface of fresh-cut potato slices.

## 1. Introduction

Potato (*Solanum tuberosum* L.) is an important food crop in the world, which contains starch, protein, amino acid, vitamins, and minerals, can provide abundant energy to the human body [1]. Generally, potato tubers are sold directly. With the demands of consumers, fresh-cut potato slices has emerged [2,3]. It is convenient to process. However, the mechanical damage caused by potato cutting results in the rupture of tissue and cells [4]. The damaged tissue undergoes browning reaction when exposed to oxygen, which causes the change of color in potato, and affects the taste and flavor. Browning is a common discoloration phenomenon in food. It is the darkening of food after processing, storage, or damage [5]. According to the mechanism of browning, it can be divided into enzymatic browning and non-enzymatic browning. The browning that occurs in fresh-cut potato slices is the enzymatic browning. When the polyphenol oxidase in potato slices comes in contact with oxygen, it accelerates the oxidation of phenolic compounds to form quinones-based polymers, which ultimately leads to the formation of pigments [6]. It causes loss of nutrients and influences the choice of consumers [7,8]. The methods to inhibit browning include heat treatment, adding inhibitors, and blocking oxygen. Browning inhibitors are usually used to inhibit browning and extend shelf life [9]. Sodium pyrosulfite is one of the commonly used browning inhibitors [10]. It is easily soluble in water and decomposes into sulfur dioxide in water [11]. The sulfur dioxide reacts with water to form sulfite. Sulfite has reducibility that acts against oxidization. Oxidization causes browning, so sulfite can inhibit the occurrence of browning. The shelf life of fresh-cut potato slices can be prolonged by immersing them in sodium pyrosulfite solution. However, as a food additive, the use of sodium pyrosulfite is strictly officially controlled. [12]. Potatoes or other fruits and vegetables immersed in sodium pyrosulfite solution are usually tested for sulfur dioxide residue on the surface of food to determine whether the added amount exceeds the standard. In China, the final residue of sulfur dioxide in fresh fruits and vegetables should be less than 0.05 g/kg [13]. If people consume food with sulfur dioxide residual amount exceeding the limit for a long time, it will cause liver and kidney damage, acute poisoning, and even death.

The common detection methods of sulfur dioxide residues in food are: hydrochloric acid rose colorimetry, distilled iodimetry, and chromatography [14,15,16]. Although these methods can accurately and quantitatively detect sulfur dioxide residues, there are some problems in the detection process, such as complex sample preparation, cumbersome experimental operation process, and expensive instruments. Therefore, a rapid detection method of sulfite residues in food is urgently needed. Near-infrared (NIR) spectroscopy has been widely used in the rapid quantitative detection of grain, fruit, vegetable, and other components. A portable NIR spectrometer can acquire the NIR spectra of the samples rapidly. Near-infrared hyperspectral imaging (NIR-HSI) technique is a rapid and nondestructive detection technology developed in recent years. It is a combination of NIR spectroscopy technique and imaging technique, which can simultaneously obtain the spectral and spatial chemical information of the detection object [17,18]. It has the advantages of simple operation, high detection efficiency, and no need for destructive pretreatment of samples. Combined with the chemometrics method, it can achieve rapid qualitative and quantitative analysis of samples. Portable NIR spectrometers are used for single-point measurement. There are relevant studies on the application and comparison of a portable NIR spectrometer and a NIR-HSI system in food detection, however there are fewer applications in the detection of sulfite residues in food [19,20,21]. Therefore, this study focuses on using a portable NIR spectrometer and a NIR-HSI system combined with chemometrics to study the feasibility of sulfur dioxide residue detection on the surface of fresh-cut potato slices immersed in different concentrations of sodium pyrosulfite solution.

The main objectives of this study are: (1) To explore the feasibility of using a portable NIR spectrometer and a NIR-HSI system to detect sulfite dioxide residues on the surface of fresh-cut potato slices immersed in different concentrations of sodium pyrosulfite solution; (2) to analyze the spectra of the samples by principal component analysis (PCA) method; (3) to recognize important wavelengths by Savitzky–Golay of second derivative (2^nd^ Der) method; (4) to establish the support vector machine (SVM) model of solution concentration grade of the samples; (5) to compare the difference between the portable NIR spectrometer and NIR-HSI system in the detection and classification of sulfite dioxide residues in the surface of fresh-cut potato slices.

## 2. Results

### 2.1. Spectral Profile

Figure 1a shows the average sample signal with standard deviation of fresh-cut potato slices immersed in different sodium pyrosulfite solution collected by the portable NIR spectrometer at the spectral range of 975–1646 nm. It could be found that the four spectral curves with the concentration of 0.05%, 0.1%, 0.3% of sodium pyrosulfite solution and distilled water had the same trend, and the four spectral curves with the concentration of 0.5%, 1%, 2%, and 3% of sodium pyrosulfite solution had the same trend. Taking the sodium pyrosulfite solution concentration of 0.05%, 0.1%, 0.3%, and the distilled water as low concentration, and taking the sodium pyrosulfite solution concentration of 0.5%, 1%, 2%, 3% as high concentration. The signal values of the samples were significantly different between the low concentration range and the high concentration range. In the range of 975–1325 nm, the signal value of fresh-cut potato slices immersed in high concentration solution were higher than that in the low concentration solution, while in the range of 1325–1646 nm, the signal value of fresh-cut potato slices immersed in high concentration solution were lower than that in low concentration solution. In the range of 1020–1120 nm and 1420–1550 nm, it could be seen that the signal value of the sample changes with the concentration of sodium pyrosulfite solution, indicating that there were differences between different samples. However, there were overlaps among the spectral curves of fresh-cut potato slices immersed in different concentrations of sodium pyrosulfite solution in general, it is necessary to further study the spectra.

Figure 1b shows the average spectral reflectance curves with standard deviation of the samples collected by the NIR-HSI system at the spectral range of 975–1646 nm. As shown in Figure 1b, at the range about 1020–1120 nm, it could be seen that with the increase of the concentration of sodium pyrosulfite solution, the value of spectral reflectance of fresh-cut potato slices increased. While at the range about 1400–1646 nm, the value of spectral reflectance decreased with the increase of the concentration of sodium pyrosulfite solution. The two band ranges showed that there were differences among the samples. It was feasible to distinguish fresh-cut potato slices immersed in different concentrations of sodium pyrosulfite solution by NIR-HSI system. However, although the spectral reflectance curves of the samples were similar in general, most of the light reflectance curves were overlapped. The spectra needed further analysis.

For portable NIR spectrometer and NIR-HSI system, they had similar spectral curve change characteristics in similar band. Because the information acquisition were in the NIR range, they were consistent in detection. Figure 1a,b show similar peak and valley ranges, respectively. The spectral peaks and valleys in the NIR spectral region are primarily related to overtones and combinations of fundamental vibrations of C-H, N-H, and O-H groups [22]. The spectral band around 1050 nm might be due to C-H stretching and C-H deformation [23]. The spectral band around 1200 nm and 1300 nm might be attributed to the second overtone of C-H stretching [22,24]. The spectral band around 1450 nm might be attributed to the combination of the first overtones of bond C-H in protein and O-H in water [25]. Although the spectral curves of the samples obtained by either the portable NIR spectrometer or the NIR-HSI system were different, there were spectral overlaps among the samples. Further spectra analysis was needed. Compared with the portable NIR spectrometer, the NIR-HSI system had less band overlapping among the samples, which might be related to the different resolution of the instruments.

### 2.2. Principal Component Analysis

Principal component analysis (PCA) was used for qualitative analysis to explore the separability among the samples. PCA was performed on the spectra obtained by the portable NIR spectrometer and NIR-HSI system in the spectral range of 975–1646 nm. The PCA scores scatter plots are shown in Figure 2. For the portable NIR spectrometer, PCA results showed that the first three principal components (PCs) (98.9% for PC1, 0.9% for PC2, and 0.1% for PC3) had explained 99.9% of the total variance of the spectra. Therefore, the distribution of all samples in the new coordinate system was determined by the first three PCs. Figure 2a shows the PCA scores scatter plots of the first three PCs by the portable NIR spectrometer. As shown in Figure 2a, there were different clustering centers among the samples. The score of PCA could reflect the intrinsic information of samples. Although the distribution of scores of the samples were overlapped, the samples immersed in the same concentration were more concentrated, forming different clustering centers. Compared with PC1 and PC2, the cumulative contribution rate of PC3 was smaller. There were more overlaps among the samples in the PCA scores scatter plots of PC1 vs. PC3 and PC2 vs. PC3. From the qualitative point of view, Figure 2a illustrated that there were differences among the samples. However, there were overlaps among many samples, and some samples are far away from the corresponding clustering centers. PCA could not classify the samples accurately. Further quantitative analysis was needed. Similarly, for the NIR-HSI system, the first three PCs (96.3% for PC1, 3.1% for PC2 and 0.5% for PC3) had explained 99.9% of the total variance of the spectra. Figure 2b shows the PCA scores scatter plots of the first three PCs by the NIR-HSI system. The samples had different cluster centers, and there were overlaps among the different samples, which required further analysis.

For the NIR-HSI system, in addition to the NIR spectra, image information could also be obtained. A hyperspectral image of one sample at each concentration of sodium pyrosulfite solution were randomly selected for analysis. The pixel-wise spectra of samples were extracted for PCA. The results showed that the cumulative contribution rate of the first three PCs was 99.63% (91.72% for PC1, 6.64% for PC2, and 1.27% for PC3), which explained most of the variance. By multiplying the scores of the first three PCs and the corresponding binary of each pixel in the binary hyperspectral image, the score images were formed by using a colorbar. Different colors represent different scores. Figure 3 shows the score images of the first three PCs of the samples. The differences among the samples were shown by different colors. The warm color (red-yellow) was the positive color scores, and the cold color (blue-green) was negative color scores. In the score image of PC1, the color score of the center position of all samples was negative, showing blue. From the center to the outside, the color score gradually became positive, showing yellow-red. Moreover, the samples with concentration of 0%, 0.05%, 0.1%, and 0.3% of sodium pyrosulfite solution showed more yellow-red, while the samples with concentration of 0.5%, 1%, 2%, and 3% showed more blue. The color score of the sample with concentration of 0% was mainly positive, and most of the color was yellow-red. It could be clearly distinguished. Similarly, for the score image of PC3, the center of all samples was blue, and the color gradually changes from the center to the outside to yellow-red. However, the samples with concentration of 0%, 0.05%, 0.1%, and 0.3% of sodium pyrosulfite solution showed more blue, and the samples with concentration of 0.5%, 1%, 2%, and 3% showed more yellow-red. The score images of PC1 and PC3 showed the color contrast of the samples with concentration of 0%, 0.05%, 0.1%, 0.3% and 0.5%, 1%, 2%, 3%, and it could classify the relative high concentration and low concentration. In the score image of PC2, except that the color score of sample with 0% concentration was almost positive, and most of the color was red, the color scores of other samples were mostly negative, and the color was green-blue. The sample with 0% concentration could be clearly distinguished. The color of the sample with 0.05% concentration was mainly blue with a large negative value, and it could be clearly distinguished. Although the PCA score images showed the differences among different samples intuitively, it was not easy to distinguish all samples accurately. It is necessary to establish a classification model for further analysis.

Compared with the portable NIR spectrometer, the NIR-HSI system could obtain image information while obtaining the NIR spectra. After acquiring the hyperspectral image by the NIR-HSI system, the pseudo-color images were formed. The difference of the samples could be intuitively displayed, which provided additional information for the samples. As shown in Figure 3, the color scores of the sample are displayed, which not only show the differences among the samples, but also shows the differences among different positions of the same sample. It was directly showed in the sample area, which was more direct than the PCA scores scatter plots in Figure 2. The information of different positions of one sample was different, and the NIR-HSI system could analyze it comprehensively. However, the Portable NIR spectrometer was a single-point measurement, which was difficult to ensure that the selected measurement point could represent the whole sample. The score image of the PCs further showed the separability among the samples.

### 2.3. SVM Models Based on Full Spectra

The results of PCA showed that there were differences among the samples. Further quantitative analysis was necessary. SVM was commonly used for the spectral data analysis models, which showed good analysis results. SVM model was used for further analysis of the samples based on the full spectra. All samples were randomly divided into a calibration set and a prediction set at a ratio of 2:1. In order to obtain the best performance, the penalty coefficient (c) and the kernel function parameter (g) were needed to be determined. The c and g can be determined through a grid search process with the search range of 10^−N^–10^N^, N = 0, 1, 2, 3...8.

For the portable NIR spectrometers, the c and g of the SVM model were 10,000,000 and 100, respectively. The classification accuracy of the calibration set was 96.25%, and the classification accuracy of the prediction set was 73.75%. Table 1. shows the confusion matrix of SVM model detected by the portable NIR spectrometer. As shown in Table 1, in the calibration set, all samples could be classified well, and the classification accuracy were over 85.00%. The accuracy of sodium pyrosulfite solution concentration of 0.50%, 1.00%, 2.00%, and 3.00% reached 100.00%, respectively. In the prediction set, except for the 0.05% concentration of sodium pyrosulfite solution, there were all samples with other concentrations misclassified, and there were many misclassified samples, leading to a decrease in classification accuracy.

For the NIR-HSI system, the c and g of the SVM model were 1,000,000 and 1000, respectively. The classification accuracy of the calibration set was 98.75%, and the classification accuracy of the prediction set was 95.00%. Table 2 shows the confusion matrix of SVM model detected by the NIR-HSI system. As shown in Table 2, in the calibration set, all samples could be classified well, and the classification accuracy were over 95.00%. In the prediction set, only the samples with the sodium pyrosulfite solution concentration of 0.50% had the classification accuracy of 80.00%, and there were a few samples misclassified with the sodium pyrosulfite solution concentration of 1.00%. The classification accuracy of the other samples was over 90.00%.

### 2.4. Important Wavelength Recognize

The spectra obtained by using the portable NIR spectrometer and NIR-HSI system to distinguish the samples were high-dimensional data, and there might be redundant information between adjacent wavelengths. Second derivative (2^nd^ Der) method is a common spectral pretreatment method, which can improve the resolution of overlapping peaks. The wavelengths corresponding to the peaks or valleys in the second derivative spectra might be related to chemical compositions, and it could be selected as important wavelengths [26,27]. Savitzky–Golay algorithm is the most commonly used algorithm in the second derivative [28]. In this study, Savitzky–Golay algorithm with the width of the moving window of three was used. Figure 4 shows the spectral curve processed by Savitzky-Golay. The wavelengths corresponding to the peaks and valleys in the curve are the important wavelengths recognized, which are shown in Table 3. As shown in Figure 4, the trend of the spectral curves of the data obtained by the portable NIR spectrometer and the NIR-HSI system after Savitzky–Golay processing was similar, and the range of corresponding important wavelengths were roughly the same. Table 3 shows the recognized important wavelengths. Total of 30 important wavelengths were recognized with the data obtained by the portable NIR spectrometer, and 32 important wavelengths were recognized with the data obtained by the NIR-HSI system. The recognized important wavelengths of spectra obtained by the portable NIR spectrometer and the NIR-HSI system were almost in the similar range.

The detection in the NIR region was based on the frequency doubling and combined absorption of molecular vibration, which was mainly related to the hydrogen-containing group X-H (X = C, N, O) [29]. Different groups and the same group have different absorption wavelengths of NIR light in different physical and chemical environments. The recognized important wavelengths played an important role in classifying the residual sulfur dioxide on the surface of fresh-cut potato slices immersed in different concentrations of sodium pyrosulfite solution. The spectral band around 985 nm might be related to the second stretching overtone of O-H [30]. The spectral band in the range of 1120–1225 nm might be related to the second overtone of C-H [31]. The spectral band in the range of 1440–1470 nm might be related to the stretching and deformation of O-H in water [32].

### 2.5. SVM Models Based on Important Wavelengths

After recognizing important wavelengths, SVM models were established for analysis based on the recognized important wavelengths.

For the portable NIR spectrometers, the c and g of the SVM model were 10,000,000 and 1000, respectively. The classification accuracy of the calibration set was 93.13%, and the classification accuracy of the prediction set was 75.00%. Table 4 shows the confusion matrix of SVM model based on the important wavelengths detected by the portable NIR spectrometer. As shown in Table 4, except for the samples immersed in 0.05% sodium pyrosulfite solution, the other samples could be classified well with the classification accuracy over 90%. In the prediction set, except for samples immersed in 0.50% sodium pyrosulfite solution, there were all samples with other concentrations misclassified. Among them, the accuracy of classification of samples immersed in distilled water and sodium pyrosulfite solution with concentration of 1.00%, 2.00%, and 3.00% were less than 70%, which were more prone to misclassification.

For the NIR-HSI system, the c and g of the SVM model were 1,000,000 and 1000, respectively. The classification accuracy of the calibration set was 92.50%, and the classification accuracy of the prediction set was 99.38%. Table 5 shows the confusion matrix of SVM model based on the important wavelengths detected by the NIR-HSI system. As shown in Table 5, in the calibration set, all samples could be classified well, and the classification accuracy were over 95.00%. In the prediction set, only the samples immersed in sodium pyrosulfite solution with concentration of 1.00% had the classification accuracy of 80.00%, the classification accuracy of the other samples was over 90.00%.

Compared with the results obtained by the SVM model based on the full spectra, the classification accuracy obtained by establishing the SVM model based on the recognized important wavelengths was slightly less than that of based on the full spectra. It showed that the recognized important wavelengths were important for the classification of the samples. Redundant information existed in the full spectra, which might cause confusion in the classification process.

According to the comparison between Table 1 and Table 2, the SVM model of NIR-HSI system could get good performance based on full spectra with the classification accuracy of calibration and prediction set of 98.75% and 95.00%, respectively. According to the comparison between Table 4 and Table 5, the SVM model of NIR-HSI system could get good performance based on recognized importance wavelengths with the classification accuracy of calibration and prediction set of 99.38% and 92.50%, respectively. Therefore, the NIR-HSI system was better than the portable NIR spectrometer. Because the classification accuracy of SVM model based on the recognized importance wavelengths was slightly less than that of based on full spectra, the recognized importance wavelengths could be used for rapid analysis.

## 3. Discussion

In this study, NIR spectra was used to analyze the sulfur dioxide residue on the surface of fresh-cut potato slices immersed in different concentrations of sodium pyrosulfite solution. It has been studied that the sulfite residue in tremella was detected by infrared spectroscopy, which showed that infrared spectroscopy was feasible for the sulfite residue in food [33]. At present, there were few studies on the detection of sulfur dioxide residues in food based on NIR spectra. In this study, we used a portable NIR spectrometer and a NIR-HSI system to acquire NIR spectra of the samples. Relevant research have shown that both imaging technique and non-imaging technique could be used to assess the quality of potatoes [22,34,35,36]. For the portable NIR spectrometer based on single-point acquisition information, it has the advantages of convenient sampling and low cost, but it was difficult to obtain comprehensive information. In the spectral acquisition process, several acquisition points of one sample were randomly selected. The spectral signal of the selected points were averaged to represent the sample as the object-wise pixel. It was difficult to ensure whether the collected signal point could represent the whole sample. The NIR-HSI system can provide spatial information of the sample while acquiring the spectral information. After obtaining the hyperspectral image, the pixel-wise spectra was extracted from each pixel of the region of one sample. The object-wise spectra and pixel-wise spectra were analyzed. Pixel-wise analysis and object-wise analysis were conducted [37]. The average of the pixel-wise spectra was calculated to represent the sample as the object-wise spectra. The spectra extracted by NIR-HSI system was more comprehensive and more representative of samples. For example, the spectral curve of samples in Figure 1b overlaps less than that in Figure 1a. Moreover, the pixel-wise spectra was used to form scores images of PCs combined with PCA. As shown in Figure 3, the differences among the samples and the differences among different positions of the same sample are intuitively displayed.

This study showed that both imaging technique and non-imaging technique could detect sulfur dioxide residue on the surface of fresh-cut potato slices. Compared with non-imaging technique, imaging technique could get more comprehensive and reliable analysis results. In the future research, the sulfur dioxide residue of fresh-cut potato slices should be measured accurately. The relationship between the sulfur dioxide residue of fresh-potato slices and the spectra needs to be further explored.

## 4. Materials and Methods

### 4.1. Materials and Sample Preparation

The potatoes (Holland fifteen) were purchased from a farm in Weifang, Shandong Province, China. Fresh potato tubers with uniform size and no obvious mechanical damage were selected. The selected potato tubers were cleaned and sliced into 3-mm thick slices along the long axis by a stainless steel slicer. Then the prepared potato slices were immersed in different concentrations of sodium pyrosulfite solution.

Sodium pyrosulfite standard was purchased from Beijing Huawei Ruike Chemical Co., Ltd. (Beijing, China) The sodium pyrosulfite standard were weighed at 0.05 g, 0.1 g, 0.3 g, 0.5 g, 1 g, 3 g, and 5 g by electronic balance, and several portions of 100 g water were weighed. The weighed sodium pyrosulfite standard and water were mixed thoroughly. The concentration of sodium pyrosulfite solution were 0.05%, 0.1%, 0.3%, 0.5%, 1%, 3%, and 5%. The concentration of 0% (distilled water) was used as control.

There were 30 potato slices set at each concentration. Seven potatoes were used to obtain 240 samples. After immersing in sodium pyrosulfite solutions of different concentrations for 15 min, the fresh-cut potato slices were dried and placed in plastic crisper at room temperature (25 °C) for 15 min before NIR spectroscopy and hyperspectral image collection.

### 4.2. Portable Near-Infrared Spectrometer

The portable NIR spectrometer we used is mainly based on a NIR module (DLP NIRscan Nano EVM, Texas Instruments, Dallas, TX, USA) for single-point measurement to acquire NIR spectra, which was consistent with the literature [20]. The wavelength range is 900–1700 nm with 228 bands. The spectral resolution is 10 nm. The spectrometer is 62 mm long, 58 mm wide, and 36 mm tall, and can be connected with the computer through USB. It is convenient to use. The spectra can be acquired by aiming the lens at the sample. When acquiring the NIR spectra, the fresh-cut potato slices were placed on a black plate. The sample window of the spectrometer was pressed against the sample, and the reflected signal of the sample was collected. For the sampling point, six times of repeated measurements were carried out and the average was calculated as the spectrum of the point. For a fresh-cut potato slice, 20 regions were randomly selected to obtain spectra, and then the average spectrum of these 20 regions was taken to represent the spectrum of the sample. A total of 240 spectra were acquired.

Because of the obvious random noise at the head and end of the spectra in the 900–1700 nm range, the 975–1646 nm range was used for further analysis. Area normalization was used to eliminate the difference of spectral intensity in the fresh-cut potato slice.

### 4.3. Near-Infrared Hyperspectral Imaging System and Spectra Extraction

#### 4.3.1. Near-Infrared Hyperspectral Imaging System

The NIR-HSI system assembled in the laboratory covering the spectral range of 874–1734 nm with 256 bands was used to collect hyperspectral image, which was consistent with that in literatures [17,38]. The spectral resolution is 5 nm. When acquiring a hyperspectral image, the black plate on which potato slices were placed was put on the image acquisition position of the NIR-HSI system. The distance between the sample and the camera lens was 12.6 cm, the exposure time of the camera was 3 ms, and the moving speed of the conveyor was 11 m/s. After the hyperspectral images were acquired, the hyperspectral images needed to be corrected. The image correction method was consistent with the literature [38]. Under the same experimental conditions of the acquisition of a sample’s hyperspectral image, a white Teflon bar with a reflectance of about 100% was placed on the sample position to obtain a white reference image. A black lens cap with a reflectance of about 0% was used to cover the lens to obtain a black reference image. Then, the raw hyperspectral images of the samples were obtained. There were 30 fresh-cut potato slices samples soaked in each concentration of sodium pyrosulfite solution, and a total of 240 hyperspectral images were acquired. Use the white reference image and the black reference image to correct the raw hyperspectral image by the following equation:(1)$$I_{c}=\frac{I_{r}-I_{d}}{I_{w}-I_{d}}$$
*I_c_* is the corrected image, *I_r_* is the original image, *I_w_* is the white reference image, and *I_d_* is the black reference image.

#### 4.3.2. Spectra Extraction

After the hyperspectral image was corrected, it was necessary to separate the sample from the background to extract the spectral information. The entire area of a complete fresh-cut potato slice was regarded as the research object and defined as the region of interest (ROI). The hyperspectral image at 1072 nm was selected as the background segmentation, because the reflectance difference between the sample and the background was largest. The background was set to 0, the sample area was set to 1, and the threshold value of binary segmentation was set to 0.1. The binary image was processed on the gray image at 1072 nm. Then the binary image was applied to the gray image of each wavelength to eliminate the background and extract the sample information. First, the pixel-wise spectra of the ROI were extracted, and then the information of each pixel in the ROI was extracted. Because of the obvious random noise at the head and end of the spectra in the 874–1734 nm range, the 975–1646 nm range was used for further analysis. Wavelet transform (WT) was used to smooth the extracted pixel-wise spectra to reduce the random noise. A Daubechies 8 wavelet function with a decomposition level of 3 was used. After WT, area normalization was used to eliminate the difference of spectral intensity in the ROI. Then the average of the pixel-wise spectra extracted from each pixel of the ROI was calculated to represent the sample reflectance spectrum, which was regarded as the object-wise spectra. The spectral extraction was conducted on MATLAB 2015a (The MathWorks, Natick, MA, USA). The pixel-wise spectra and object-wise spectra were both used for analysis.

### 4.4. Data analysis

#### 4.4.1. Principal Component Analysis

Principal component analysis (PCA) transforms the original data variables into new orthogonal variables through linear transformation, and uses the new orthogonal variables to express the data characteristics of the original variables as much as possible [39,40]. The new orthogonal variables are called principal components (PCs). The number of PCs can be determined by calculating the cumulative contribution rate of PCs. Usually the first few PCs contain most useful information and explain most of the total variance [41].

For PCA of hyperspectral image, there are object-wise analysis and pixel-wise analysis [37]. Object-wise analysis is to use the average spectrum of one sample instead of individual pixels for PCA to get the scatter plot. Samples of different PCs are scattered in different areas, and then the differences among the samples were explored [42]. Pixel-wise analysis is based on individual pixels of the hyperspectral image to obtain the score of individual pixels in the hyperspectral image of each PC to form a score visualization image, which can intuitively observe the differences among the samples [37]. In this study, PCA was performed on the spectra collected by portable NIR spectrometer and the object-wise spectra and pixel-wise spectra extracted from hyperspectral images obtained by NIR-HSI system.

#### 4.4.2. Important Wavelength Recognition Method

Second derivative (2^nd^ Der) method was used to recognize the important wavelength. The application of 2^nd^ Der method in spectral analysis is helpful to eliminate the linear trend and increase the difference between spectra [43]. The spectra processed by the 2^nd^ Der method could suppress the spectral noise and highlight the spectral peaks. The wavelengths corresponding to the peaks or valleys in the second derivative spectra might be related to chemical compositions, and it could be selected as important wavelengths [26,27]. Savitzky–Golay algorithm is the most commonly used algorithm in the second derivative. It carries out smooth filtering, which can improve the smoothness of the spectra and reduce the noise interference. The effect of Savitzky–Golay varies with the selected window width [28]. In this study, the width of the moving window was set to three.

#### 4.4.3. Classification Analysis Method

Support vector machine (SVM) was used to classify the residual sulfur dioxide on the surface of fresh-cut potato slices immersed in different concentrations of sodium pyrosulfite solution. SVM is a nonlinear modeling method [44]. It can separate the samples of different categories by realizing the optimal classification hyperplanes, which has the advantages of high prediction ability and low classification error rate [45]. Kernel function is an important part of SVM model. Radial basis function is a widely used kernel function, and it can obtain the best performance of the model by determining the penalty coefficient (c) and the kernel function parameter (g).

To conduct SVM model, it is necessary to divide calibration set and prediction set. In this study, the samples were randomly divided into calibration set and prediction set at a ratio of 2:1. There were no duplicate samples in calibration set and prediction set. At the corresponding concentration of sodium pyrosulfite solution, the second sample of every three samples was selected into the prediction set, and the remaining two samples were selected into the calibration set. Therefore, there were 160 samples in the calibration set and 80 samples in the prediction set.

#### 4.4.4. Software

The hyperspectral image was analyzed by ENVI4.7 (ITT Visual Information Solutions, Boulder, CO, USA). ENVI provides a visual environment for hyperspectral images. It was used to define the sample area as ROI for subsequent analysis. MATLAB 2015a (The MathWorks, Natick, MA, USA) is a practical mathematical operation software. The PCA function and libSVM 3,1 toolbox in MATLAB were used to conduct PCA and SVM model. The Unscrambler^®^ 10.1 (Camo as, Oslo, Norway) is a multivariable data analysis software. It has the Savitzky–Golay processing option that can directly process the spectra. All figures were realized by Origin 2018 (Origin lab Corporation, Northampton, Ma, USA).

## 5. Conclusions

The portable NIR spectrometer and NIR-HSI system were used to study the feasibility of sulfur dioxide residue detection on the surface of fresh-cut potato slices immersed in different concentrations of sodium pyrosulfite solution. Although portable NIR spectrometer was convenient and low cost, the NIR-HSI system could provide more comprehensive information and acquired more satisfactory results. The score images of first three PCs showed the difference between the samples. The SVM model based on full spectra had the classification accuracy of calibration set and prediction set as 98.75% and 95.00% respectively. The SVM model based on the recognized important wavelengths by Savitzky–Golay had the classification accuracy of calibration set and prediction set as 99.38% and 92.50% respectively. The SVM model of the NIR-HSI system based on full spectra had better results. The NIR-HSI system was able to classify better among the different concentrations. Moreover, the classification accuracy of SVM model based on the recognized importance wavelengths was slightly less than that of based on full spectra. The recognized important wavelengths could be used for rapid analysis. This study provided a reliable method for the detection of sulfur dioxide residue on the surface of fresh-cut potato slices. In the future research, the relationship between the actual sulfur dioxide residue on the surface of fresh-cut potato slices and the obtained spectra should be further explored. The other vegetables and fruits could be considered.

## Figures and Tables

**Figure 1 molecules-25-01651-f001:**
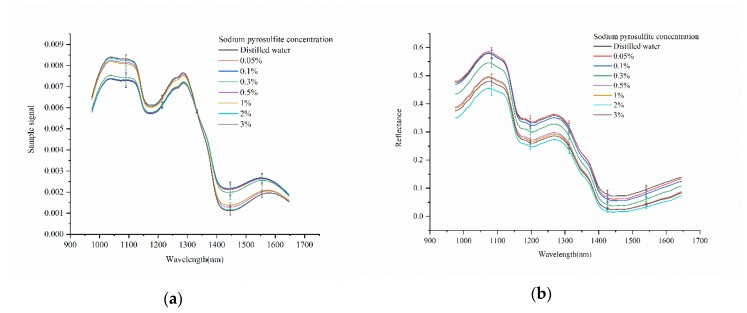
The average spectra with standard deviation of fresh-cut potato slices immersed in different sodium pyrosulfite solution collected by (**a**) portable NIR spectrometer and (**b**) NIR-HSI system.

**Figure 2 molecules-25-01651-f002:**
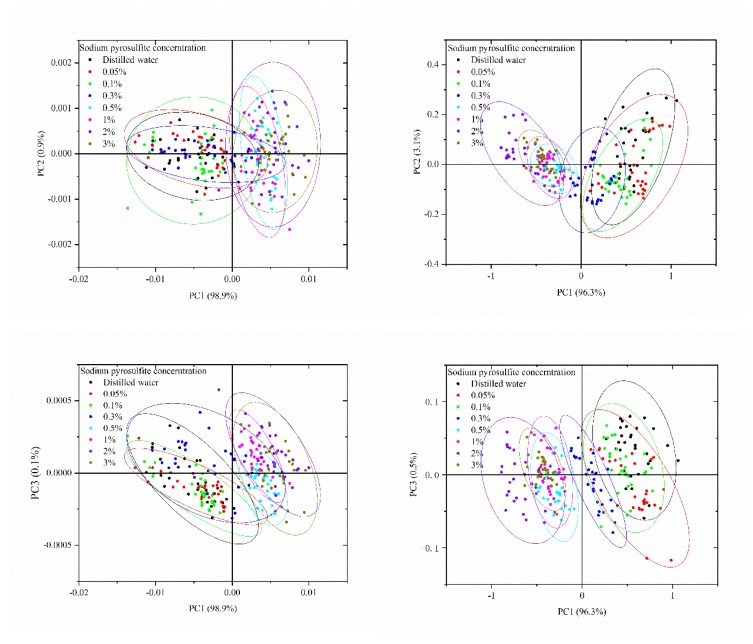
Principal component analysis (PCA) scores scatter plots of the first three principal components (PCs): (**a**) portable NIR spectrometer and (**b**) NIR-HSI system.

**Figure 3 molecules-25-01651-f003:**
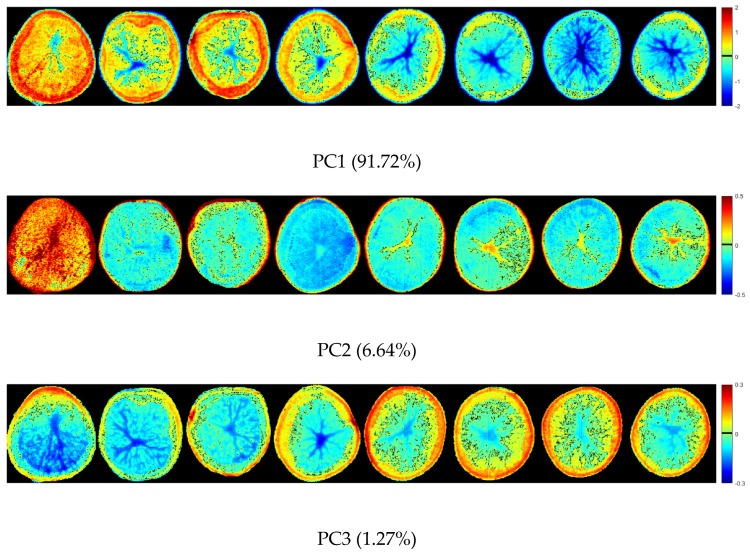
Scores images of the first three PCs (the concentration of the sodium pyrosulfite solution were: 0% (distilled water), 0.05%, 0.1%, 0.3%, 0.5%, 1%, 2% and 3%).

**Figure 4 molecules-25-01651-f004:**
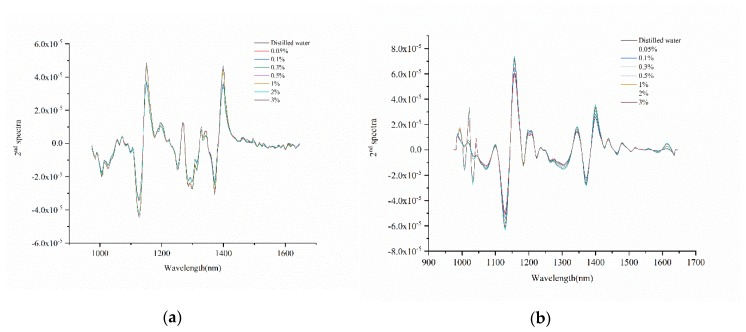
Spectral curve processed by Savitzky–Golay: (**a**) portable NIR spectrometer and (**b**) NIR-HSI system.

**Table 1 molecules-25-01651-t001:** Confusion matrix of SVM model based on full spectra for classification the sulfur dioxide residues on the surface of fresh-cut potato slices immersed in different concentrations of sodium pyrosulfite solution by the portable NIR spectrometers.

		0% ^1^	0.05%	0.10%	0.30%	0.50%	1.00%	2.00%	3.00%	Accuracy
Cal.^2^	0%	19	1	0	0	0	0	0	0	95.00%
	0.05%	2	17	1	0	0	0	0	0	85.00%
	0.10%	0	0	19	1	0	0	0	0	95.00%
	0.30%	1	0	0	19	0	0	0	0	95.00%
	0.50%	0	0	0	0	20	0	0	0	100.00%
	1.00%	0	0	0	0	0	20	0	0	100.00%
	2.00%	0	0	0	0	0	0	20	0	100.00%
	3.00%	0	0	0	0	0	0	0	20	100.00%
	Total									96.25%
Pre.^3^	0**%**	6	4	0	0	0	0	0	0	60.00%
	0.05%	0	10	0	0	0	0	0	0	100.00%
	0.10%	1	1	8	0	0	0	0	0	80.00%
	0.30%	0	2	0	8	0	0	0	0	80.00%
	0.50%	0	0	0	1	9	0	0	0	90.00%
	1.00%	0	0	0	1	3	6	0	0	60.00%
	2.00%	0	1	0	0	0	0	5	4	50.00%
	3.00%	0	0	0	0	1	0	2	7	70.00%
	Total									73.75%

^1^ 0% represents the distilled water. ^2^ Cal. represents the calibration set. ^3^ Pre. represents the prediction set.

**Table 2 molecules-25-01651-t002:** Confusion matrix of support vector machine (SVM) model based on full spectra for classification the sulfur dioxide residues on the surface of fresh-cut potato slices immersed in different concentrations of sodium pyrosulfite solution by the NIR-HSI system.

		0% ^1^	0.05%	0.10%	0.30%	0.50%	1.00%	2.00%	3.00%	Accuracy
Cal.^2^	0%	19	1	0	0	0	0	0	0	95.00%
	0.05%	0	20	0	0	0	0	0	0	100.00%
	0.10%	0	0	20	0	0	0	0	0	100.00%
	0.30%	0	0	0	20	0	0	0	0	100.00%
	0.50%	0	0	0	0	19	1	0	0	95.00%
	1.00%	0	0	0	0	0	20	0	0	100.00%
	2.00%	0	0	0	0	0	0	20	0	100.00%
	3.00%	0	0	0	0	0	0	0	20	100.00%
	Total									98.75%
Pre.^3^	0**%**	10	0	0	0	0	0	0	0	100.00%
	0.05%	0	10	0	0	0	0	0	0	100.00%
	0.10%	0	1	9	0	0	0	0	0	90.00%
	0.30%	0	0	0	10	0	0	0	0	100.00%
	0.50%	0	0	0	0	8	2	0	0	80.00%
	1.00%	0	0	0	0	1	9	0	0	90.00%
	2.00%	0	0	0	0	0	0	10	0	100.00%
	3.00%	0	0	0	0	0	0	0	10	100.00%
	Total									95.00%

^1^ 0% represents the distilled water. ^2^ Cal. represents the calibration set. ^3^ Pre. represents the prediction set.

**Table 3 molecules-25-01651-t003:** The important wavelengths recognized by Savitzky–Golay.

	Important Wavelengths(nm)
Portable NIR Spectrometer	913, 941, 962, 985, 1005, 1013, 1028, 1055, 1070, 1101, 1128, 1150, 1176, 1195, 1224, 1250, 1268, 1293, 1307, 1329, 1343, 1371, 1399, 1452, 1499, 1602 1633, 1646, 1667, 1692
NIR-HSI System	985, 992, 1009, 1022, 1032, 1042, 1072, 1099, 1130, 1156, 1183, 1200, 1207, 1224, 1237, 1247, 1264, 1274, 1304, 1345, 1372, 1399, 1426, 1440, 1467, 1480, 1507, 1521, 1558, 1588, 1612, 1636

**Table 4 molecules-25-01651-t004:** Confusion matrix of SVM model based on the important wavelengths for classification the samples by the portable NIR spectrometers.

		0% ^1^	0.05%	0.10%	0.30%	0.50%	1.00%	2.00%	3.00%	Accuracy
Cal.^2^	0%	19	1	0	0	0	0	0	0	95.00%
	0.05%	2	14	4	0	0	0	0	0	70.00%
	0.10%	0	2	18	0	0	0	0	0	90.00%
	0.30%	1	0	0	19	0	0	0	0	95.00%
	0.50%	0	0	0	0	20	0	0	0	100.00%
	1.00%	0	0	0	0	0	20	0	0	100.00%
	2.00%	0	0	0	0	0	0	19	1	95.00%
	3.00%	0	0	0	0	0	0	0	20	100.00%
	Total									93.13%
Pre.^3^	0**%**	6	3	0	1	0	0	0	0	60.00%
	0.05%	0	9	1	0	0	0	0	0	90.00%
	0.10%	0	1	9	0	0	0	0	0	90.00%
	0.30%	0	2	0	8	0	0	0	0	80.00%
	0.50%	0	0	0	0	10	0	0	0	100.00%
	1.00%	0	0	0	1	2	7	0	0	70.00%
	2.00%	0	0	1	0	0	2	5	2	50.00%
	3.00%	0	0	0	0	3	0	1	6	60.00%
	Total									75.00%

^1^ 0% represents the distilled water. ^2^ Cal. represents the calibration set. ^3^ Pre. represents the prediction set.

**Table 5 molecules-25-01651-t005:** Confusion matrix of SVM model based on the important wavelengths for classification the samples by the NIR-HSI system.

		0% ^1^	0.05%	0.10%	0.30%	0.50%	1.00%	2.00%	3.00%	Accuracy
Cal.^2^	0%	19	1	0	0	0	0	0	0	95.00%
	0.05%	0	20	0	0	0	0	0	0	100.00%
	0.10%	0	0	20	0	0	0	0	0	100.00%
	0.30%	0	0	0	20	0	0	0	0	100.00%
	0.50%	0	0	0	0	20	0	0	0	100.00%
	1.00%	0	0	0	0	0	20	0	0	100.00%
	2.00%	0	0	0	0	0	0	20	0	100.00%
	3.00%	0	0	0	0	0	0	0	20	100.00%
	Total									99.38%
Pre.^3^	0**%**	10	0	0	0	0	0	0	0	100.00%
	0.05%	1	8	1	0	0	0	0	0	80.00%
	0.10%	0	1	9	0	0	0	0	0	90.00%
	0.30%	0	0	0	10	0	0	0	0	100.00%
	0.50%	0	0	0	0	9	1	0	0	90.00%
	1.00%	0	0	0	0	1	9	0	0	90.00%
	2.00%	0	0	0	0	0	0	10	0	100.00%
	3.00%	0	0	0	0	0	1	0	9	90.00%
	Total									92.50%

^1^ 0% represents the distilled water. ^2^ Cal. represents the calibration set. ^3^ Pre. represents the prediction set.

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
