# Peer review of "Detection of Sulfite Dioxide Residue on the Surface of Fresh-Cut Potato Slices Using Near-Infrared Hyperspectral Imaging System and Portable Near-Infrared Spectrometer"

_molecules, 2020, doi:10.3390/molecules25071651_

Round 1
Reviewer 1 Report
The manuscript tries to identify and quantify the presence of sulfur dioxide residue on the surface of freshly ​​potato slices. Sulfur dioxide is produced from the sodium pyrosulfite used to prevent browning of potato slices. To highlight the presence of this product harmful to humans, the authors propose two alternative methods to conventional methods of analysis: NIR spectroscopy and hyperspectral images in the near infrared.
The manuscript is interesting, original, novel and important for the journal. However, as structured and drafted the manuscript is not understood. Thus, for example, some abbreviations that appear in section 2 (PCA, PCs, SVM, etc.) are explained in section 3. If abbreviations are to be used, they will have to be explained the first time they are named.
Lines 83-85 are not Results, it is rather Materials and Methods. Similar paragraphs are repeated numerous times throughout the entire manuscript to introduce the different sections. Authors should eliminate or modify them so as not to repeat themselves so much.
The authors only comment on the first two graphs in Figure 2. The results shown in the other 4 graphs PC1 vs PC3 and PC2 vs PC3 should also be commented in the text.
Lines 183-186. How is the random separation of the calibration and prediction set performed?
The parameters ‘c’ and ‘g’ are named for the first time on line 187, but they are not defined until line 362. The authors should solve this problem that is repeated for many abbreviations, parameters, methods, materials,….
Section 3.1. lines 287-290. How are NIR spectra taken from potatoes? Does the spectrophotometer have fiber optics, integrating sphere or a sample compartment?
Section 3.2. How are the measurements performed, in reflectance? Authors should specify
The authors, although they have put a section called Results and Discussion, do not discuss the results obtained and should do so to.
Section 5. Lines 371-389. The authors have not written true Conclusions of the work, this is more a summary than conclusions. Authors should make an effort to draw a conclusion from their work. They have done a good research, they have good results, but they have not drawn conclusions. They should decide which methods have been better between NIRS and NIR-HSI or, at least, give advantages and disadvantages of each of them. In the same way, they should conclude which method of data processing has given better results and is able to discriminate better between the different concentrations.
The order of the sections of the article prevents us from understanding it. To understand the article, I had to read section 3 first and then 2. It seems that the authors have first written section 3, then 2 and then reversed the order. The authors must solve this problem. The authors can write the two sections again so that the text makes sense.
Author Response
Response to Reviewer 1 Comments
Dear reviewer:
Thank you very much for your comments on our manuscript. The listed comments are all very valuable and helpful for revising and improving our paper, as well as the important guiding significance to our research. We have studied the comments carefully and have modified the manuscript that we hope to meet with approval (marked in red). Please see the revised manuscript at the attachment. The responses to the reviewer’s comments are as follow:
Point 1: The manuscript is interesting, original, novel and important for the journal. However, as structured and drafted the manuscript is not understood. Thus, for example, some abbreviations that appear in section 2 (PCA, PCs, SVM, etc.) are explained in section 3. If abbreviations are to be used, they will have to be explained the first time they are named.

Response 1: We are sorry for our carelessness. We have checked the abbreviations used in the manuscript, and the abbreviations were explained the first time they were named. PCA was explained in Line 81, PCs was explained in Line 134 and SVM was explained in Line 83.
Point 2: Lines 83-85 are not Results, it is rather Materials and Methods. Similar paragraphs are repeated numerous times throughout the entire manuscript to introduce the different sections. Authors should eliminate or modify them so as not to repeat themselves so much.
Response 2: We are sorry for our carelessness. The repeated sentence in the manuscript have been eliminated.
Point 3: The authors only comment on the first two graphs in Figure 2. The results shown in the other 4 graphs PC1 vs PC3 and PC2 vs PC3 should also be commented in the text.
Response 3: Thanks for your comment. We have added more information in Lines 135-142, “Therefore, the distribution of all samples in the new coordinate system was determined by the first three PCs. Figure 2(a) shows the PCA scores scatter plots of the first three PCs by the portable NIR spectrometer. As shown in Figure 2(a), there were different clustering centers among the samples. The score of PCA could reflect the intrinsic information of samples. Although the distribution of scores of the samples were overlapped, the samples immersed in the same concentration were more concentrated, forming different clustering centers. Compared with PC1 and PC2, the cumulative contribution rate of PC3 was smaller. There were more overlaps among the samples in the PCA scores scatter plots of PC1 vs PC3 and PC2 vs PC3.”
Point 4: Lines 183-186. How is the random separation of the calibration and prediction set performed?
Response 4: Thanks for your comment. The separation of the calibration and prediction set was introduced in detail in section 4.3.3. We have added more information in Lines 426-431, “To conduct SVM model, it is necessary to divide calibration set and prediction set. In this study, the samples were randomly divided into calibration set and prediction set at a ratio of 2: 1. There were no duplicate samples in calibration set and prediction set. At the corresponding concentration of sodium pyrosulfite solution, the second sample of every three samples was selected into the prediction set, and the remaining two samples were selected into the calibration set. Therefore, there were 160 samples in the calibration set and 80 samples in the prediction set.”
Point 5: The parameters ‘c’ and ‘g’ are named for the first time on line 187, but they are not defined until line 362. The authors should solve this problem that is repeated for many abbreviations, parameters, methods, materials,….
Response 5: We are sorry for our carelessness. We have added more information in Lines 199-201, “In order to obtain the best performance, the penalty coefficient (c) and the kernel function parameter (g) were needed to be determined. The c and g can be determined through a grid search process with the search range is 10-N-10N, N=0, 1, 2, 3...8.”
Point 6: Section 3.1. lines 287-290. How are NIR spectra taken from potatoes? Does the spectrophotometer have fiber optics, integrating sphere or a sample compartment?
Section 3.2. How are the measurements performed, in reflectance? Authors should specify
Response 6: Thanks for your comment.
- About acquiring near-infrared spectra from potatoes, we have added relevant information in Lines 347-349, “When acquiring the NIR spectra, the fresh-cut potato slices were placed on a black plate. The sample window of the spectrometer was pressed against the sample, and the reflected signal of the sample was collected.”, and Lines 360-361, “the black plate on which potato slices placed was put on the image acquisition position of the NIR-HSI system.”
- The spectrometer do not have fiber optics, integrating sphere and a sample compartment.
- For the measurement performed, we have added more information in section 4.2, “The spectrometer is 62mm long, 58mm wide and 36mm tall, and can be connected with the computer through USB. It is convenient to use. The spectra can be acquired by aiming the lens at the sample. When acquiring the NIR spectra, the fresh-cut potato slices were placed on a black plate. The sample window of the spectrometer was attached to the sampling point of the sample, and the reflected signal of the sample was collected.”
Point 7: The authors, although they have put a section called Results and Discussion, do not discuss the results obtained and should do so to.
Response 7: Thanks for your comment. We have added the new section 3. Discussion. “In this study, NIR spectra was used to analyze the sulfur dioxide residue on the surface of fresh-cut potato slices immersed in different concentrations of sodium pyrosulfite solution. It has been studied that the sulfite residue in tremella was detected by infrared spectroscopy, which showed that infrared spectroscopy was feasible for the sulfite residue in food [1]. At present, the detection of sulfur dioxide residues in food based on NIR spectra was relatively few. In this study, we used a portable NIR spectrometer and a NIR-HSI system to acquire NIR spectra of the samples. Relevant research have shown that both imaging technique and non-imaging technique could be used to detect the quality of potatoes [2-5]. For the portable NIR spectrometer based on single-point acquisition information, it has the advantages of convenient sampling and low cost, but it was difficult to obtain comprehensive information. In the spectral acquisition process, several acquisition points of one sample were randomly selected. The spectral signal of the selected points were averaged to represent the sample as the object-wise pixel. It was difficult to ensure that the collected signal point could represent the whole sample. The NIR-HSI system can provide spatial information of the sample while acquiring the spectral information. After obtaining the hyperspectral image, the pixel-wise spectra was extracted from each pixel of the region of one sample. The object-wise spectra and pixel-wise spectra could be analyzed. Pixel-wise analysis and object-wise analysis could be conducted [6]. The average of the pixel-wise spectra was calculated to represent the sample as the object-wise spectra. The spectra extracted by NIR-HSI system was more comprehensive and more representative of samples. For example, the spectral curve of samples in Figure 1 (b) overlaps less than that in (a). Moreover, the pixel-wise spectra could be used to form scores images of PCs combined with PCA. As shown in Figure 3, the differences among the samples and the differences among different positions of the same sample are intuitively displayed.
This study showed that both imaging technique and non-imaging technique could detect sulfur dioxide residue on the surface of fresh-cut potato slices. Compared with non-imaging technique, imaging technique could get more comprehensive and reliable analysis results. In the future research, the sulfur dioxide residue of fresh-cut potato slices should be measured accurately. The relationship between the sulfur dioxide residue of fresh-potato slices and the spectra needed to further explore.”
- Yuan, K.; Wang, X.; Zhang, L.; Gu, D.; Guo, X.; Fan, Z., Rapid and quantitative determination of sulfite residue in tremella based on IR spectroscopy and PLS. Food & Machinery 2017, 33, (6), 64-67,134. https://doi.org/10.13652/j.issn.1003-5788.2017.06.013.
- Su, W.; Sun, D., Potential of hyperspectral imaging for visual authentication of sliced organic potatoes from potato and sweet potato tubers and rapid grading of the tubers according to moisture proportion. Electro. Agr. 2016, 125, 113-124. https://doi.org/10.1016/j.compag.2016.04.034
- Chao, K.; Chin, B. A.; Cho, B.; Kim, M. S.; De Biasio, M.; Arnold, T., Study of near-infrared imaging spectroscopy for the inspection of peeled potato tubers. In Sensing for Agriculture and Food Quality and Safety X, SPIE: Orlando, United States, 2018.
- Escuredo, O.; Seijo-Rodríguez, A.; Inmaculada González-Martín, M.; Shantal Rodríguez-Flores, M.; Carmen Seijo, M., Potential of near infrared spectroscopy for predicting the physicochemical properties on potato flesh. J. 2018, 141, 451-457. https://doi.org/https://doi.org/10.1016/j.microc.2018.06.008.
- Liang, P.-S.; Haff, R. P.; Hua, S.-S. T.; Munyaneza, J. E.; Mustafa, T.; Sarreal, S. B. L., Nondestructive detection of zebra chip disease in potatoes using near-infrared spectroscopy. Eng. 2018, 166, 161-169. https://doi.org/https://doi.org/10.1016/j.biosystemseng.2017.11.019.
- Feng, L.; Zhu, S.; Zhang, C.; Bao, Y.; Feng, X.; He, Y. Identification of Maize Kernel Vigor under Different Accelerated Aging Times Using Hyperspectral Imaging. Molecules 2018, 23. https://doi.org/10.3390/molecules23123078.
Point 8: Section 5. Lines 371-389. The authors have not written true Conclusions of the work, this is more a summary than conclusions. Authors should make an effort to draw a conclusion from their work. They have done a good research, they have good results, but they have not drawn conclusions. They should decide which methods have been better between NIRS and NIR-HSI or, at least, give advantages and disadvantages of each of them. In the same way, they should conclude which method of data processing has given better results and is able to discriminate better between the different concentrations.
Response 8: Thanks for your comment. We have revised the section 5: “The portable NIR spectrometer and NIR-HSI system were used to study the feasibility of sulfur dioxide residue detection on the surface of fresh-cut potato slices immersed in different concentrations of sodium pyrosulfite solution. Although portable NIR spectrometer was convenient and low cost, the NIR-HSI system could provide more comprehensive information and acquired more satisfactory results. The score images of first three PCs could showed the difference among the samples. The SVM model based on full spectra had the classification accuracy of calibration set and prediction set with 98.75% and 95.00% respectively. The SVM model based on the recognized important wavelengths by Savitzky-Golay had the classification accuracy of calibration set and prediction set with 99.38% and 92.50% respectively. The SVM model of the NIR-HSI system based on full spectra had better results. The NIR-HSI system was able to classify better among the different concentrations. Moreover, the classification accuracy of SVM model based on the recognized importance wavelengths was slightly less than that of based on full spectra. The recognized importance wavelengths could be used for rapid analysis. This study provided a reliable method for the detection of sulfur dioxide residue on the surface of fresh-cut potato slices. In the future research, the relationship between the actual sulfur dioxide residue on the surface of fresh-cut potato slices and the obtained spectra should be further explored. The other vegetables and fruits could be considered.”
Point 9: The order of the sections of the article prevents us from understanding it. To understand the article, I had to read section 3 first and then 2. It seems that the authors have first written section 3, then 2 and then reversed the order. The authors must solve this problem. The authors can write the two sections again so that the text makes sense.
Response 9: Thanks for your comment. However, due to the requirements of the journal guide for authors and the article template, the results and discussion should be before the materials and methods. We must comply with the requirements of journals. To make the content more coherent, we have added some simple introduction of the theory in the corresponding parts of section 2.Results.
Thank you again for your comment. We hope our corrections would meet with your approval.
Reviewer 2 Report
Title: (first reading): clear.
(after having read the article): Can be shortened:
'Study on' unnecessary (just delete).
It is also unclear why to emphasize 'portable',
the imaging system is not portable.
Abstract:
Second derivative method: Savitzky-Golay more specific (and well known).
Introduction:
Line 33: 'Solanum tuberosum' in italics font.
Line 34: 'heat': mean 'energy'
Line 35: 'sold directly in the market' what does that mean? 'market square', 'retail sales', or what?
Lines 35-36: 'With the ...' sentence is unclear: the word ordering is strange: rewrite / ask English expert help.
Line 38: 'exposed to air': should be 'exposed to oxygen'?? (nitrogen etc does not cause browning?)
BTW, the authors could shortly explain what is browing and why it happens and how it can be prevented.
Line 42: 'Sodium pyrosulfite is a kind of sulfite.' YES, it is, but saying that does not give any further information: so just delete this short sentence.
Line 44: 'Sulfite has reducibility...' YES, it acts against oxididation which is the cause of 'browning'.
Line 45: '...obviously after..' authors mean: ' .. by ..'?
Line 45: 'soaking in' better: 'soaking them in'.
Line 46: '..is limited [10].' better something like '..is strictly officially controlled...'
Line 49: '..be less than 0.05%.' Give also the unit: weight percentage?
Lines 52-53: avoid 'and so on' better: '..in food include ....'.
(applies also to the next sentence on Lines 53-55).
Line 58: (grammar: a when telling the first time): 'A portable...'
Lines 61-62: '..internal chemical information and external characteristic information of the detection object[]' the authors mean ' .. spatial chemical information of the object...'
Lines 64-65: Why use singular when not describing one device but a class of devices??? 'Portable NIR spectrophotometers are used for ... while NIR-HSI systems are able to obtain spatially more comprehensible ...'
Lines 68-71: long sentence that could be made more clear by word ordering: In English the most important is put to the beginning of the sentence (paragraph etc).
Line 73: '...detect and classify...' what is meant by 'classify'?
After having read the paper, there is one phrase that repeats over and over again:
'fresh-cut potato slices immersed in different concentrations of sodium pyrosulfite solution'
Please, think how to delete most of its occurrences by a shorter phrase and/or suitable pronouns. (pronouns are like hyperlinks)
2. Results and discussion
This title just after one page introduction is confusing.
Please tell first the 'theory' i.e. scientific background and
then your experimental set up and then results and discussion.
Lines 83-85: Tell you measurement protocol as a separate chapter.
Line 83: 'gradient': Can't see gradient, just a few concentration values in different measurements.
Lines 86-93: Tell about the lab protocol used to make the solutions and the chemicals used.
BTW, NIR spectra tend to be non-linear. How about here?
Figure 1: The spectra should be quite similar. Any idea why they seem to be somewhat different. Sensor sensitivity or???
Lines 118-124: Make a more specific analysis.
2.2. PCA:
Please tell about the methods used (not just the software brand).
Line 133: 99.9% total variance seems quite high. How could you check this result?
Lines 160-161: color coding: the authors should tell how the color coding is related to the more traditional presentation of results.
Lines 172-178: Could the authors be more specific (numerical) in their analysis.
2.3. SVM:
The authors should tell more about the SVM used: what is the motivation to use the values of the parameters that wer given in line 187?
2.4. Important wavelengths
Please tell in theory part about the S-G and its parameters and the statistical properties of the spectra analyzed. Give also reference to S-G.
Table 3: Rank the peaks.
3. Material and methods
This should be earlier (before results).
Lines 313-314: Explain how the images were corrected.
BTW, how about calibration of the devices and and methods used?
3.4.4. Software
The brands does not much tell about the exact methods used.
Please specify the methods.
5. Conclusions:
Please think about calibration of the devices.
Reference:
Be consistent with reference style.
Round 2
Reviewer 1 Report
There is an error in the quote:
Escuredo, O.; Seijo-Rodríguez, A.; Inmaculada González-Martín, M.; Shantal Rodríguez-Flores, M.; Carmen Seijo, M. Potential of near infrared spectroscopy for predicting the physicochemical properties on potato flesh. Microchem. J. 2018, 141, 451-457. https://doi.org/https://doi.org/10.1016/j.microc.2018.06.008.
This should be:
Escuredo, O.; Seijo-Rodríguez, A.; González-Martín, M.I.; Rodríguez-Flores, M.S.; Seijo, M.C. Potential of near infrared spectroscopy for predicting the physicochemical properties on potato flesh. Microchem. J. 2018, 141, 451-457. https://doi.org/https://doi.org/10.1016/j.microc.2018.06.008.
Inmaculada, Shantal and Carmen are first name, no last name
Author Response
Dear reviewer:
Thank you very much for your comments on our manuscript. The listed comments are all very valuable and helpful for revising and improving our paper, as well as the important guiding significance to our research. We have studied the comments carefully and have modified the manuscript that we hope to meet with approval (marked in red). The responses to the reviewer’s comments are as follow:
Point 1: There is an error in the quote:
Escuredo, O.; Seijo-Rodríguez, A.; Inmaculada González-Martín, M.; Shantal Rodríguez-Flores, M.; Carmen Seijo, M. Potential of near infrared spectroscopy for predicting the physicochemical properties on potato flesh. Microchem. J. 2018, 141, 451-457. https://doi.org/https://doi.org/10.1016/j.microc.2018.06.008.

This should be:
Escuredo, O.; Seijo-Rodríguez, A.; González-Martín, M.I.; Rodríguez-Flores, M.S.; Seijo, M.C. Potential of near infrared spectroscopy for predicting the physicochemical properties on potato flesh. Microchem. J. 2018, 141, 451-457. https://doi.org/https://doi.org/10.1016/j.microc.2018.06.008.
Inmaculada, Shantal and Carmen are first name, no last name
Response 1: We are sorry for our carelessness. We have revised the quote in the manuscript.
Thank you again for your comment. We hope our corrections would meet with your approval.